# Overexpression of DGAT2 Stimulates Lipid Droplet Formation and Triacylglycerol Accumulation in Bovine Satellite Cells

**DOI:** 10.3390/ani12141847

**Published:** 2022-07-20

**Authors:** Jun-Fang Zhang, Seong-Ho Choi, Qiang Li, Ying Wang, Bin Sun, Lin Tang, En-Ze Wang, Huan Hua, Xiang-Zi Li

**Affiliations:** 1Engineering Research Center of North-East Cold Region Beef Cattle Science & Technology Innovation, Ministry of Education, Department of Animal Science, Yanbian University, Yanji 133002, China; junfangzhang0613@163.com (J.-F.Z.); liqiang8589@ybu.edu.cn (Q.L.); 13180937133@163.com (Y.W.); a2653152414@163.com (B.S.); t1422906017@163.com (L.T.); wez19980804@163.com (E.-Z.W.); hh390584731@163.com (H.H.); 2Department of Animal Science, Chungbuk National University, Cheongju 28644, Korea; seongho@cbnu.ac.kr

**Keywords:** DGAT2, satellite cell, intramuscular fat, triacylglycerol, overexpression, transcriptome

## Abstract

**Simple Summary:**

Triacylglycerols are the major component of intramuscular fat, and the final step in their biosynthesis is catalyzed by DGAT2. Bovine skeletal muscle satellite cells were infected with the overexpression adenovirus Ad-DGAT2 and interfering adenovirus sh-DGAT2. The results showed that overexpression of DGAT2 upregulated the expression of genes involved in lipid accumulation and adipogenesis and increased cellular triacylglycerol content. The differentially expressed genes were mainly enriched in the PPAR signaling, glycerolipid metabolism, fatty acid biosynthesis, and AMPK signaling pathways. These results highlight the important regulatory role of DGAT2 during adipogenic transdifferentiation of BSCs and the complexity of intramuscular adipogenesis, as well as providing a theoretical basis for producing high marbling content beef.

**Abstract:**

Intramuscular fat (IMF) is closely related to the tenderness, juiciness, and flavor of beef, and is an important indicator for beef quality assessment internationally. The main components of skeletal intramuscular fat (IMF) are phospholipids and triacylglycerols (TAG), and the final step of TAG biosynthesis is catalyzed by diacylglycerol acyltransferase 2 (DGAT2). To explore the effect of DGAT2 on the differentiation of bovine muscle satellite cells (BSCs) and its role in the signaling pathway related to lipid metabolism, the adenovirus overexpression and interference vector of the DGAT2 gene was constructed in this study, and the overexpression adenovirus Ad-DGAT2 and interfering adenovirus sh-DGAT2 were used to infect BSCs. Overexpression of DGAT2 resulted in a significant increase in the contents of TAG and ADP, and the mRNA and protein expression levels of PPARγ, C/EBPα, and SREBF1 (*p* < 0.05). Interfering with the expression of DGAT2 reduced the intracellular TAG content and lipid droplet accumulation. Furthermore, the mRNA and protein expression levels of PPARγ, C/EBPα, and SREBF1 (*p* < 0.05) were significantly downregulated. Transcriptome sequencing showed that a total of 598 differentially expressed genes (DEGs) were screened in BSCs infected with Ad-DGAT2, and these DEGs included 292 upregulated genes and 306 downregulated genes. A total of 49 DEGs were screened in BSCs infected with sh-DGAT2, and these DEGs included 25 upregulated and 24 downregulated genes. KEGG enrichment analysis showed that the DEGs, after overexpression of DGAT2, were mainly enriched in the PPAR signaling pathway, and the fat digestion and absorption, glycerophospholipid metabolism, fatty acid biosynthesis, and AMPK signaling pathways. The DEGs obtained after interfering with DGAT2 were mainly enriched in the metabolic pathways, such as the PPAR signaling pathway and PI3K/AKT signaling pathway. In summary, our study demonstrated that the lipid droplet formation, TAG accumulation, and adipogenic gene expression in BSCs overexpressing DGAT2 were higher than those in the control cells. These results highlight the important role of DGAT2 in regulating BSCs during adipogenic transdifferentiation and underscore the complexity of intramuscular adipogenesis.

## 1. Introduction

The fat content in skeletal muscle is closely related to the sensory and edible quality of beef and has a great influence on the flavor, color, juiciness, and tenderness of meat. This includes intramuscular fat (IMF) located within the perimysium, intermuscular fat (INTMF) located between muscle bundles, and intramyocellular lipids (IMCL) [1]. IMF content is an important indicator of meat quality. The main components of IMF are phospholipids and triglycerides, the content of which is mainly determined by number and size, and which is associated with the proliferation and differentiation capacity of intramuscular adipocytes, which provide a site for the subsequent “marbling” of fat deposits. Skeletal muscle PDGFRα + FAPs cell populations and satellite cell populations serve as the main sources of intramuscular adipocytes, which are key targets in the regulation of intramuscular adipocyte development and “marbling” fat deposition [2]. Satellite cells are skeletal muscle progenitor/stem cells that reside between the basement membrane and plasma membranes of skeletal muscle fibers in vivo and can give rise to muscle cells and adipocytes. Furthermore, the proliferation and adipogenic differentiation potential of skeletal muscle satellite cells are closely related to IMF deposition [3,4]. IMCL are mainly stored in muscle fibers, in the form of lipid droplets. The components of lipid droplets in skeletal muscle are esterified lipids, mainly triacylglycerol (TAG), followed by cholesterol and diacylglycerol (DAG) [5,6]. There are two pathways for TAG synthesis: the phosphoglycerol pathway (Kennedy pathway), and the monoacylglycerol (MG) pathway [7]. The phosphoglycerol pathway functions in most cells, while the MG pathway functions only in specialized cells. The final reaction of the two pathways uses diacylglycerol and fatty acid acyl to synthesize TAG [8], and the final step of its synthesis is catalyzed by fatty acyl-CoA-diacylglycerol acyltransferases (DGAT). These enzymes are referred to as DGAT1 or DGAT2, based on their order of recognition.

Studies have shown that DGAT1-mediated TAG synthesis not only plays a role in energy storage, but also protects adipocytes from lipotoxicity during lipolysis in mice lacking DGAT1 in their adipocytes [9]. Mice lacking DGAT1 survive and are metabolically healthy, with ~50% reduction in TG storage in adipose tissue [10]. In sharp contrast to DGAT1 knockout mice, mice lacking DGAT2 died shortly after birth. Importantly, DGAT2 knockout mice had significantly reduced triglyceride (TG) storage (less than 10% of normal levels) [8,11]. DGAT2 differs from DGAT1 in that it has a single membrane-embedded hairpin and its catalytic domain is located on the cytoplasmic side of the endoplasmic reticulum membrane. The activity of DGAT2 is largely dependent on TG synthesis. In addition, DGAT2 can also relocate around the lipid droplets required for local TG synthesis [12,13]. Whether the DGAT2 gene plays a similar role in lipid infiltration in bovine skeletal muscle tissue remains unclear. We hypothesized that DGAT2 encodes proteins with critical roles and induces adipogenic transdifferentiation of BSCs. In this study, adenovirus overexpression and interference vectors of the DGAT2 gene were constructed, and Ad-DGAT2 overexpressing adenovirus and sh-DGAT2 interference adenovirus were obtained. The effects of the DGAT2 gene on the expression of fat-related genes at the mRNA and protein levels were analyzed, to explore the effects of DGAT2 on BSC differentiation and lipid metabolism-related signaling pathways in cattle.

## 2. Materials and Methods

### 2.1. Ethics Statement

The Animal Care and Use Committee (approval code YBU-20160303) of Yanbian University (Yanji, Jilin, China) approved all procedures involving the use of live cattle.

### 2.2. Adenovirus Generation and Overexpression of DGAT2

The cDNA sequence of Yanbian cattle DGAT2 (Accession No. NM_205793.2) was subcloned into the ADV1 shuttle plasmid vector between EcoRI and BamHI, to construct a recombinant shuttle plasmid containing the target gene, and then the recombinant shuttle plasmid was combined with the backbone plasmid pGP-Ad-Co-transfected 293A cells with Pac Vector (recombinant shuttle plasmid and packaging plasmid were constructed by GenePharma Biotech, Shanghai, China), using a new adenovirus backbone gene (with deletion of left ITR and packaging signal and E1 sequence; homologous recombination in bacteria at the pAdEasy step was not necessary) to generate the adenovirus pAd-DGAT2. A recombinant adenovirus with green fluorescent protein (Ad-GFP) was used as a negative control. The virus titer was detected using the micropan-cytopathic method (refer to the method of the Institute of Genetics of Fudan University, Shanghai, China), and the Ad-DGAT2 and Ad-NC adenovirus titers were 1 × 109 PFU/mL.

### 2.3. Design of shRNAs and Recombinant Adenovirus Production

Three shRNAs targeting DGAT2 were designed and detected using BLAST. Based on the position of the shRNA targeting the target gene CDS region sequence, they were named DGAT2-shRNA-108, DGAT2-shRNA-320, and DGAT2-shRNA-687. In addition, shRNA-NC sequences that did not target any genes were designated as the control group. The target sequences are shown in Table 1, with sequence information.

The shRNA duplex was then cloned into the corresponding site of the adenovirus shuttle plasmid ADV1 BamHI/EcoRI to generate recombinant ADV1/U6/CMV shRNA containing the shRNA expression cassette, in which the shRNA expression was driven by the U6 promoter (PU6). The resulting recombinant shuttle plasmid was co-transfected into 293A cells, along with the backbone plasmid pGP-Ad-Pac vector. sh-NC was constructed by GenePharma Biotech (Shanghai, China).

### 2.4. Optimization of Optimal Conditions for Adenoviral Overexpression/Interference of the DGAT2 Gene

The day before the test, 5 × 103 BSCs were inoculated into each well of a 96-well plate, at a total volume of 100 µL. Interference pretests were divided into two groups. The first group was normally infected, and the virus was added directly to the complete medium. The second group was treated with 5 µg/mL polybrene (GenePharma Biotech, Shanghai, China) at the time of infection, and each group had a different multiplicity of infection (MOI) gradient. Before infection, the cell medium was changed, and the adenovirus stock solution was serially diluted (stock solution, 10-fold dilution, 100-fold dilution, 1000-fold dilution, 10,000-fold dilution). These dilutions were used to infect BSCs. Subsequently, the cells were cultured for 24 h, and the medium was then replaced with fresh medium. After 48 h infection, expression was observed by fluorescence and the infection parameters of the target cells were confirmed. Infection efficiency was evaluated by detecting the expression of the target gene using real-time PCR.

### 2.5. Cell Culture and Treatment

BSCs were isolated from the semimembranosus muscle of 8-day-old Yanbian cattle using pronase (10165921001, Roche, Basel, Switzerland) digestion, as described previously [14]. Details of the BSC culture were also been described in our previous studies [14,15]. Briefly, the BSCs were incubated in DMEM (Gibco, Thermo Fisher Scientific, Waltham, MA, USA) containing 10% fetal bovine serum (FBS) (Gibco, Thermo Fisher Scientific, Waltham, MA, USA) in a humidified incubator at 37 °C and 5% CO_2_. When the BSCs were about 80% confluent, they were transfected with adenovirus supernatants of recombinant adenovirus expressing DGAT2 (Ad-DGAT2), Ad-NC, recombinant adenovirus interfering with DGAT2 (sh-DGAT2), or sh-NC. After 24 h of adenovirus infection, differentiation was induced using oleic acid (OA), in differentiation medium containing 5% FBS, 100 µM OA, and DMEM. Transfected BSCs were collected after 96 h of culture for total RNA extraction, protein extraction, and triglyceride determination.

### 2.6. RNA Extraction and Quantitative Real-Time PCR

Total RNA was extracted from cells incubated for 96 h using TRIzol reagent (Thermo Fisher Scientific), according to the manufacturer’s instructions. The extracted total RNA was assayed for integrity using a NanoDrop ND-100 spectrophotometer (2000C, Thermo Fisher Scientific) and further verified using 1% agarose gel electrophoresis. cDNA was synthesized using a FastKing One Step Kit (Tiangen Biotech, Beijing, China), according to the manufacturer’s protocol. Quantitative real-time PCR was performed using SYBR Green (SuperReal PreMix Plus; Tiangen Biotech, Beijing, China) on an Agilent Mx3000/5p Real-Time PCR Detection System (Agilent Mx3000/5p; Agilent Technologies, Santa Clara, CA, USA), according to the manufacturer’s instructions. PCR amplification was performed at 95 °C for 15 min, followed by 40 cycles at 95 °C for 15 s, 60 °C for 30 s, and 72 °C for 30 s. The dissociation protocol used incremental temperatures of 95 °C for 15 s and 65 °C for 5 s.

Genes involved in the evaluation included the genes associated with adipogenesis: peroxisome proliferator-activated receptor gamma (PPARγ), CCAAT/enhancer binding protein alpha (C/EBPα), CCAAT/enhancer binding protein beta (C/EBPβ); genes for de novo fatty acid synthesis: acetyl-CoA carboxyl acylase alpha (ACACA), fatty acid synthase (FASN), and desaturation-related genes stearoyl-CoA desaturase (SCD); TAG synthesis-related genes: glycerol-3-phosphate acyltransferase 4 (GPAT4), 3-phosphoglycerol acyltransferase (GPAM), monoacylglycerol acyltransferase 1 (MGAT1), diacylglycerol acyltransferase 1 (DGAT1), and diacylglycerol acyltransferase 2 (DGAT2); paired box 7 (Pax7), myogenic differentiation antigen (MYOD), myogenin (MYOG), and myofactor-4 (MRF4) genes to determine the extent of myogenesis in cultured BSCs.

Gene expression values were calculated using the relative quantification (2^−ΔΔCt^) method [16], and glyceraldehyde-3-phosphate dehydrogenase (GAPDH) was used as an endogenous reference for normalization of the target RNA profiles; the primer sequences are shown in Table 2.

### 2.7. Oil Red O Staining

Oil Red O staining was used to confirm the degree of lipid droplet accumulation during adipogenic transdifferentiation of BSCs. BSCs were stained with Oil Red O, and the specific staining procedure was carried out using an Oil Red O Staining Kit (G1262, Solarbio, Beijing, China), according to the instructions provided by the manufacturer. Briefly, the cells were fixed in 4% paraformaldehyde, washed with 60% (*v/v*) aqueous isopropanol, incubated with Oil Red O staining solution for 30 min, and then stained with hematoxylin for 1 min. The stained cells were observed using an inverted microscope (IX-73, Olympus, Tokyo, Japan). Adipocytes were identified based on the presence of red-stained lipid droplets.

### 2.8. Cellular TAG Assay

The TAG concentration in cells was measured using a TAG assay kit (Applygen Technologies, Beijing, China), according to the manufacturer’s instructions. Briefly, adenovirus-infected BSCs were washed three times with PBS, lysed with RIPA buffer (10 min) (R0010, Solarbio, Beijing, China), and cell lysates were extracted. TAG content was measured at 570 nm by enzymatic colorimetry, using a microplate reader (iMark, BIO-RAD, Hercules, CA, USA).

### 2.9. Adiponectin Assay in Culture Medium

Adiponectin, after adenovirus infection and induction of differentiation for 96 h, was analyzed using a bovine adiponectin ELISA Kit (Mlbio, Shanghai, China), and the analysis was performed according to the manufacturer’s protocol. Adiponectin concentrations were measured at 450 nm using a microplate reader (iMark, BIO-RAD, Hercules, CA, USA), and a standard curve was generated in the same assay using reference standards of known concentrations of adiponectin.

### 2.10. Western Blotting

Transfected cells were lysed in RIPA buffer containing 1% phenylmethylsulfonyl fluoride (PMSF). A BCA protein kit (Pierce, Thermo Fisher Scientific Inc., Waltham, MA, USA) was used to determine the total protein concentration. Protein samples were separated using 10% SDS-PAGE and transferred onto PVDF membranes (Millipore, Billerica, MA, USA). The membranes were incubated overnight at 4 °C with the following antibodies: anti-DGAT1 (ab189994, Abcam, Cambridge, UK), anti-DGAT2 (ab59493, Abcam, Cambridge, UK), anti-PPARγ (bs-4509R, Bioss, Beijing, China), anti-C/EBPα (bs-1630R, Bioss, Beijing, China), anti-SREBF1(bs-1402R, Bioss, Beijing, China), anti-Pax7 (ab61067, Abcam, Cambridge, UK), anti-MYOD (bs-2442R, Bioss, Beijing, China), and anti-MYOG (bs-3550R, Bioss, Beijing, China). Thereafter, the membranes were washed and incubated with horseradish peroxidase-conjugated goat anti-mouse or goat anti-rabbit IgG (bs-0295G; Bioss, Beijing, China) for 1 h. β-actin (ab8226, Abcam, Cambridge, UK) was used as an endogenous control. A grayscale intensity analysis was performed using ImageJ software (NIH).

### 2.11. Transcriptome Sequencing (RNA-Seq) Research

BSCs were treated with adenovirus Ad-NC, Ad-DGAT2, sh-NC, or sh-DGAT2, and induced to differentiate for 96 h. Trizol was used to extract total RNA from the cells. The concentration of total RNA was quantified using a NanoDrop-NC2000 spectrophotometer (Thermo Scientific, Shanghai, China), and its purity was assessed using 1% (*w*/*v*) agarose gel electrophoresis. Samples with an RIN value of >7 were used for library preparation. RNA purification, reverse transcription, library construction, and sequencing were performed by Shanghai Personal Biotechnology Co. Ltd. (Shanghai, China).

The sequencing results from Illumina^®^ HiSeq TM 2000 were raw reads. The sequencing data contained low-quality reads with adapters, including reads with an average quality score lower than Q20, and needed to be further filtered. Ideally, the distribution of reads should be uniform across all the expressed genes. Gene coverage uniformity shows the sequence coverage in the 5′ to 3′ region of all genes in each sample and was used to assess the uniformity of the sequencing results. We used RSeQC to analyze the expression level saturation, to assess whether the amount of data measured was sufficient to correctly calculate gene expression levels. We used HTSeq to statistically compare the read count value of each gene to the original expression of the gene and normalized the expression using FPKM. Pearson’s correlation coefficient was used to represent the correlation of gene expression levels between the samples. The closer the correlation coefficient is to 1, the higher the similarity of expression patterns between samples. DESeq was used for differential analysis of gene expression, and the conditions for screening differentially expressed genes (DEGs) were as follows: expression difference fold |log2FoldChange| > 1, significant *p*-value < 0.05, and the R command p.adjust was used to correct multiple comparisons.

### 2.12. Gene Ontology (GO) and Kyoto Encyclopedia of Genes and Genomes (KEGG) Pathway Analysis of DEGs

First, the DEGs were mapped to the database established by the Gene Ontology Consortium (Gene Ontology, http://geneontology.org/, accessed on 20 April 2022). We then used topGO to perform GO enrichment analysis, using the differential genes annotated by GO terms to analyze the difference of each term. The gene list was determined and the number of genes was calculated. Then, the *p*-value was calculated by the hypergeometric distribution method (the criterion for significant enrichment is *p*-value < 0.05), and the GO terms with significant enrichment of differential genes compared with the whole genome background were identified. Thus, the main biological functions of the differential genes were determined. GO covers three aspects: the molecular function of genes (MF), cellular component function (CC), and biological process (BP).

### 2.13. KEGG Enrichment Analysis

KEGG (http://www.kegg.jp/, accessed on 20 April 2022) enrichment analysis was used to identify pathways that were significantly enriched in differentially expressed transcripts compared to the whole genome. We used clusterProfiler to perform a KEGG enrichment analysis, and used the differential genes annotated by the KEGG pathway to determine the gene list and number of genes for each pathway. Then, the *p*-value was calculated by the hypergeometric distribution method (the criterion for significant enrichment was *p*-value < 0.05), and the KEGG pathway in which the differential genes were significantly enriched compared with the whole genome background was determined, to identify the main biological functions exercised by the differential genes.

### 2.14. Statistical Analysis

The results are expressed as the mean ± SEM of three independent experiments. Data evaluation and statistical analysis were performed using SPSS 17.0 and GraphPad Prism 6.07 (GraphPad Software, La Jolla, CA, USA) software. Student’s *t*-test was used to compare the differences between the two groups. Differences between more than two groups were assessed using one-way ANOVA. Statistical significance was set at *p* < 0.05.

## 3. Results

### 3.1. Detection of Optimal MOI Value and Infection Efficiency in Yanbian Bovine Skeletal Muscle Satellite Cells Infected with Adenovirus Overexpressing or Interfering with DGAT2

As shown in Figure 1, when the polybrene concentration was 5 μg/mL and the adenovirus dilution factor was 10^−4^ (MOI = 100), Ad-DGAT2, Ad-NC (A), sh-DGAT2, and sh-NC (B) adenovirus-infected cells had no lesions and a good morphology, and all expressed green fluorescence.

As shown in Figure 2, real-time PCR detected DGAT2 gene expression and found that in the culture containing Ad-DGAT2, the mRNA expression of DGAT2 was almost 1700-fold higher than that in culture containing Ad-NC (*p* < 0.01) (Figure 2A). Compared with the sh-NC control, DGAT2-shRNA-320 successfully mediated the interference of the DGAT2 gene in BSCs, and the interference efficiency reached 67.4% (Figure 2B). Therefore, the optimal MOI for subsequent experiments was determined to be 100.

### 3.2. Overexpression of DGAT2 Increases TAG Accumulation, ADP Content, and Lipid Droplet Formation in BSCs Undergoing Adipogenic Transdifferentiation

BSCs were infected with recombinant adenovirus expressing DGAT2 (Ad-DGAT2) and control (adenovirus containing green fluorescent protein; Ad-NC) for 24 h and then differentiated by treatment with OA for 96 h. Lipid droplet measurement and TAG and ADP analyses were performed. As shown in Figure 3A, lipid droplet accumulation was significantly induced in BSCs transfected with Ad-DGAT2 compared with that in BSCs transfected with Ad-NC. Relative to Ad-NC, the overexpression of DGAT2 significantly increased cellular TAG accumulation (Figure 3B), and the adiponectin content in the medium also significantly increased (*p* < 0.05; Figure 3C).

### 3.3. Interference of DGAT2 Affects TAG Accumulation, ADP Content, and Lipid Droplet Formation in BSCs Undergoing Adipogenic Transdifferentiation

BSCs were infected with recombinant adenovirus containing shRNA fragments targeting the DGAT2 gene (DGAT2-shRNA-320) and a control (adenovirus containing shRNA-NC sequences not targeting any gene; sh-NC) for 24 h, and differentiation was induced via treatment with OA for 96 h. Oil Red O staining and TAG and ADP analyses were performed. The results are shown in Figure 4A. Lipid droplet formation was significantly inhibited in BSCs transfected with DGAT2-shRNA-320 compared with that in BSCs transfected with sh-NC controls. Interference with DGAT2 significantly reduced cellular TAG accumulation relative to sh-NC (Figure 4B), and the adiponectin content in the medium was also significantly reduced (*p* < 0.05; Figure 4C).

### 3.4. The Expression and Function of DGAT2 Overexpression in BSCs Adipogenic Differentiation

As shown in Figure 5A, compared to the control Ad-NC, the mRNA expression of PPARγ, C/EBPβ, SREBF1, and FABP4 was significantly upregulated in Ad-DGAT2-infected cells, and the expression of C/EBPα was also significantly upregulated (*p* < 0.05). Compared with Ad-NC, overexpression of DGAT2 significantly upregulated the expression of the triacylglycerol synthesis-related genes GPAT4 and LPIN1, as well as DGAT1 and GPAM (*p* < 0.05) (Figure 5B). Compared to the Ad-NC control, overexpression of DGAT2 significantly upregulated the mRNA expression of ACACA, FASN, and SCD (*p* < 0.05; Figure 5C). As shown in Figure 5D, the overexpression of DGAT2 significantly suppressed the expression of PAX7, MYOD1, MYOG, and MYR4 in BSCs (*p* < 0.05). The expression levels of these proteins are shown in Figure 5E (Appendix A).

### 3.5. Effects of Interference with DGAT2 Expression on BSCs Undergoing Adipogenic Differentiation

Compared with the mRNA expression in sh-NC-infected BSCs, the mRNA expression of PPARγ, C/EBPα, C/EBPβ, and SREBF1 was significantly decreased in BSCs infected with sh-DGAT2 during OA-induced adipogenic transdifferentiation (*p* < 0.05; Figure 6A). As shown in Figure 6B, compared to the sh-NC control, interference with DGAT2 expression caused a significant increase in the expression of DGAT1 and decreased the mRNA expression of GPAT4 and LPIN1 (*p* < 0.05), but there was no significant difference in the expression of MGAT1 (*p* > 0.05). Compared with sh-NC, DGAT2 interference significantly suppressed the expression of ACACA and SCD (*p* < 0.05) but had no significant effect on FASN (Figure 6C) (*p* > 0.05). As shown in Figure 6D, compared to sh-NC, sh-DGAT2 infection of BSCs significantly inhibited the mRNA expression of PAX7, MYOD1, MYOG, and MYR4 (*p* < 0.05; Figure 6D). The expression levels of the related proteins are shown in Figure 6E (Appendix A).

### 3.6. DEG Screening in BSCs Infected with Ad-DGAT2/sh-DGAT2

The quality of the RNA-seq data was assessed, sequencing data were filtered, and the composition ratio of each part of the original data was analyzed. According to the sequencing quality evaluation results, the high-quality clean reads of both groups were >91.00% (Appendix A). In this study, the genome of bovine *Bos taurus* (bovine) (https://www.ncbi.nlm.nih.gov/genome/?term=Bos+taurus, accessed on 20 April 2022) was used as the reference genome, and the sequencing data were compared with the reference genome (Appendix A). Figure 7 shows the sequence of the gene coverage uniformity (Figure 7A) and FPKM density distribution (Figure 7B). A total of 598 DEGS were screened in BSCs infected with Ad-DGAT2, and these DEGs included 292 upregulated genes and 306 downregulated genes. A total of 49 DEGS were screened in BSCs infected with sh-DGAT2, and these DEGs included 25 upregulated and 24 downregulated genes (Figure 7C).

### 3.7. GO Functional Classification

As shown in Figure 8A, the top 20 GO terms enriched by 598 DEGs after overexpression of DGAT2 were mainly related to two categories: biological processes, such as multicellular organismal process, cell differentiation, cellular development process, and striated muscle cell differentiation; and cellular compounds, such as extracellular matrix and myofibril. A GO functional classification of the DEGs obtained after infection with sh-DGAT2 revealed enrichment in three categories: biological processes, cellular compounds, and molecular functions. As shown in Figure 8A, the number of upregulated and downregulated genes in the basolateral plasma membrane was highest in the cellular compound category. In addition, DEGs were significantly enriched in the categories of biological processes, such as cellular responses to triglycerides. The molecular function category was mainly related to enzyme activity.

### 3.8. KEGG Enrichment Analysis

Pathway analysis of the DEGs was performed to understand the pathways and molecular interactions related to DGAT2. As shown in Figure 9A, a KEGG enrichment analysis of DEGs after overexpression of DGAT2 in BSCs undergoing differentiation revealed that DEGs were mainly enriched in the PPAR signaling pathway; fat digestion and absorption, glycerolipid metabolism, fatty acid biosynthesis, and AMPK signaling pathways; and glycerophospholipid metabolism, biosynthesis of unsaturated fatty acids, and other signaling pathways.

There were 10 DEGs enriched in the PPAR signaling pathway, including six upregulated genes: ACSL3, FADS2, SCD, HMGCS1, FABP3, and FABP7, and four downregulated genes: ANGPTL4, CD36, PLIN4, and APOA1(Figure 9B). There were six DEGs enriched in the glycerolipid metabolism pathway, all of which were upregulated: DGAT2, LPIN1, LIPG, PNPLA3, MOGAT1, and LOC101905992(Figure 9C). There were seven DEGs enriched in the AMPK signaling pathway, including six upregulated genes, SCD, FASN, SREBF1, ACACA, SLC2A4, and FBP2, and one downregulated gene, CD36 (Figure 9D).

A pathway enrichment significance analysis of DEGs in the BSCs after infection with sh-DGAT2 was performed using the KEGG database, and 20 significant pathways were found to be enriched, including the phospholipase D signaling pathway, phosphoinositide 3-Kinase (PI3K)/protein kinase B (Akt) signaling pathway, PPAR signaling pathway, and neutrophil extracellular trap formation (Figure 10A). Two DEGs in the PPAR signaling pathway, PLIN4 and SLC27A5, were downregulated (Figure 10B). KEGG enrichment analysis helped us further understand the biological functions of the DEGs.

## 4. Discussion

The proportion of saturated, monounsaturated, and polyunsaturated fatty acids in beef has a significant impact on human health [17]. Studies have shown that the high saturated fatty acid (SFA) component of beef is associated with increased blood cholesterol levels and that consumption of meat containing high SFA and trans fatty acids decreases HDL levels and increases TAG and the atherosclerosis index (LDL/HDL) compared with consumption of meat containing high monounsaturated fatty acid (MUFA) [18,19]. Intracellular fatty acids can be imported from extracellular sources or synthesized by cells [20]. DGAT catalyzes the final step of TAG formation through acylation of diacylglycerols (DAGs) [21]. Muscle satellite cells exhibit multipotent mesenchymal stem cell activity and are capable of forming osteocytes, adipocytes, and myocytes [22]. Our previous study showed that the lipid droplet area was significantly increased and TAG accumulation was promoted in BSCs differentiated by palmitoleic acid induction [23]. Belal et al. [24] showed that long-chain fatty acids (LCFAs) induced cytosolic TAG accumulation in satellite cells and significantly enhanced DGAT2 mRNA expression; however, of the two DGAT isoforms expressed in BSCs, only DGAT2 significantly responded to LCFA treatment, indicating that DGAT2 plays a critical role in TAG accumulation in cattle.

The study of mouse DGAT2 by McFie et al. [25] showed that DGAT2 is part of a multimeric complex composed of several DGAT2 subunits, and the interaction of DGAT2 with lipid droplets depends on the C-terminus of DGAT2. When cells were incubated with OA to stimulate TAG synthesis, DGAT2 mutants, in which the C-terminal region was truncated or specific regions were deleted, were unable to colocalize with lipid droplets. Their findings demonstrated that DGAT2 can catalyze TAG synthesis and promote TAG storage in cytosolic lipid droplets, regardless of its localization in the endoplasmic reticulum. Zhang et al. [26] studied the fatty acid composition of 3T3-L1 preadipocytes overexpressing DGAT2 and found that a higher proportion of MUFA C16:1 and C18:1 were derived from de novo fatty acid synthesis, rather than from the absorption of specific fatty acids in the culture medium. This is consistent with our findings that the mRNA expression levels of ACACA, FASN, and SCD, which are involved in de novo fatty acid synthesis, were significantly higher in BSCs overexpressing DGAT2 than in control cells. However, the expression of genes such as ACACA was significantly suppressed in sh-DGAT2-tranfected BSCs. This study aimed to determine the mechanism of DGAT2 in the adipogenic transdifferentiation of BSCs, to provide a theoretical basis for the actual production of beef rich in UFA.

Overexpression of DGAT2 in mouse glycolytic muscle increased the content of TAG, ceramide, and unsaturated long-chain fatty acyl-CoA in adult mice, mediating skeletal muscle-specific lipid deposition [27]. Our study showed that overexpression of DGAT2 significantly increased the TAG content in BSCs and ADP content in the medium. The mRNA expression of adipogenesis-related genes PPARγ, C/EBPβ, SREBF1, and FABP4 was also significantly increased. SCD1 and DGAT2 are located in close proximity to the endoplasmic reticulum, and SCD has been proposed to mediate the conversion of de novo MUFA to DGAT2 as a substrate for TAG synthesis [28]. Solé et al. [29] showed that DGAT2 preferentially uptakes shorter rather than longer-chain fatty acids as a substrate, especially if they are monounsaturated. Numerous studies have shown that low DGAT2 expression leads to reduced TAG content. DGAT2-knockout mice, (DGAT2(-/-)), are fat-deficient and die shortly after birth, due to a severe reduction in energy metabolism substrates [11]. The siRNA-mediated downregulation of goose DGAT2 reduces the mRNA levels of adipogenesis-related genes [30]. Our results showed that after interfering with the expression of DGAT2, the formation of lipid droplets in the cells was significantly inhibited, intracellular TAG content was significantly reduced, and the adiponectin content in the medium was also significantly reduced. The mRNA expression of lipogenesis-related genes PPARγ, SREBF1, and SCD, and lipid metabolism genes GPAT4 and LPIN1, was also significantly downregulated.

We focused on identifying key genes involved in adipogenesis and lipid metabolism. Transcriptome sequencing after overexpression of DGAT2 revealed 598 DEGs. The top 20 GO terms enriched by these 598 DEGs included muscle system process, cell differentiation, and cell development process. KEGG enrichment analysis showed that these DEGs were mainly involved in PPAR signaling, glycerophospholipid metabolism, and AMPK signaling pathways. As a known transcription factor, PPARγ can regulate the expression of ACS, FABP3, FABP4, and others [31]. The upregulated DEGs GPAT4, GPAM, and LPIN1 in the glycerophospholipid metabolic pathway have been identified as important regulators of lipid deposition [32,33]. AMPK signaling plays a crucial role in systemic metabolic pathways by regulating energy homeostasis and cellular lipid metabolism [34,35,36]. Overexpression of DGAT2 upregulated the AMPK downstream lipid synthesis-related genes SREBF1, FAS, ACC1, and SCD1, to regulate lipogenesis.

Analysis of the enriched GO terms derived from the 49 DEGs obtained after disrupting DGAT2 revealed that the DEGs were significantly enriched in GO terms, such as cellular response to triglycerides and cellular compounds, and were mainly involved in metabolic pathways, such as the PPAR signaling and PI3K/Akt signaling pathways. The PI3K/Akt signaling pathway is involved in various cellular functions, such as adipocyte differentiation. Previous studies have shown that inhibition of the PI3K/Akt signaling pathway results in reduced obesity and improved metabolic health [37]. After interference with the expression of DGAT2, the expression of SREBF1 and SCD was downregulated. In this study, we found that regulating the expression of DGAT2 positively and cooperatively regulates the mRNA expression of SREBF1 and SCD, and alteration of PPAR and SREBPs levels by up- or downregulation of DGAT2 may imply a role for DGAT2 in regulating BSC differentiation into adipocytes.

There are three types of differentiated adipogenic cells, including white, brown, and beige adipocytes, and the three types are morphologically difficult to recognize; however, a common feature of all adipoblasts is the expression of FABP4 [38,39,40]. A study by Schubert et al. [41] on the differentiation capacity of satellite cells from the rotator cuff revealed that all FABP4+ cells were also Pax7-tdTomato+, suggesting that these adipocytes originate from a myogenic progenitor population. Furthermore, in adipogenic media, satellite cells have reduced myogenic differentiation potential, and the expression of the differentiated myocyte marker MRF4 is decreased; accompanied by a significant upregulation of the adipogenesis transcription factor PPAR and FABP4 expression, and increased adipogenic differentiation potential. In our study, enhancing or interfering with the expression of DGAT2 significantly inhibited the mRNA and protein expression levels of the myogenesis-related genes PAX7, MYOD1, MYOG, and MYR4. There is reciprocal inhibition between Pax7 and muscle regulatory factors (MRFs), and Pax7 null mice exhibited reduced muscle growth, marked muscle atrophy, and extremely deficient muscle regeneration after acute injury [42,43]. This suggests that the myogenic differentiation of BSCs was inhibited. Therefore, the upregulation of these adipogenic proteins and the downregulation of myoblast differentiation proteins promote OA-induced transdifferentiation of BSCs into adipocytes, inhibit their myogenic differentiation ability, and promote lipid accumulation in cells.

## 5. Conclusions

The accumulation of TAG and adipogenic gene expression in BSCs overexpressing DGAT2 were higher than those in the control cells in this study. Interference of DGAT2 expression inhibited lipid droplet formation in OA-induced differentiated BSCs. DGAT2 participated in the PPAR signaling pathway, glycerophospholipid metabolism pathway, AMPK signaling pathway, and PI3K/Akt signaling pathway to coordinately regulate lipid production, by controlling the transcription of certain genes in BSCs.

## Figures and Tables

**Figure 1 animals-12-01847-f001:**
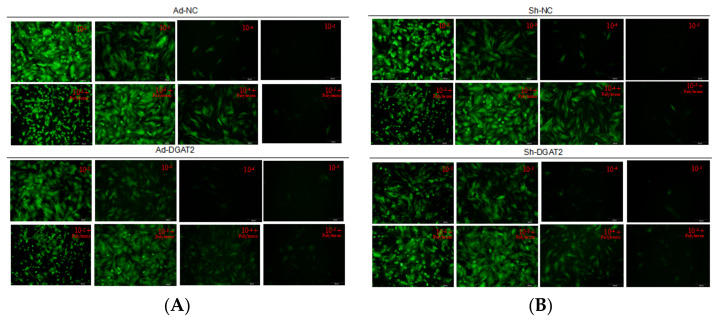
Green fluorescent expression of high titer adenovirus Ad-NC, Ad-DGAT2 (**A**) and sh-NC, sh-DGAT2 (**B**) infected bovine skeletal muscle satellite cells after 48 h. Scale bars = 200 µm.

**Figure 2 animals-12-01847-f002:**
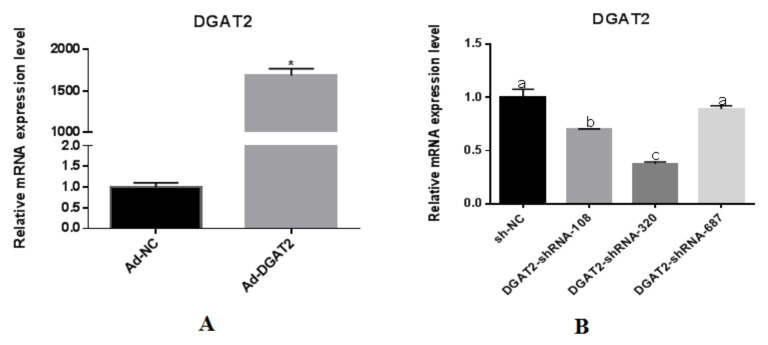
Effects of overexpression (**A**) and interference (**B**) of DGAT2 gene on mRNA expression levels in bovine skeletal muscle satellite cells. Values are presented as means ± SEM. The different letters (a–c) represent significant differences (*p* < 0.05) in gene expression. * *p* < 0.05 compared with the control (Ad-NC/sh-NC).

**Figure 3 animals-12-01847-f003:**
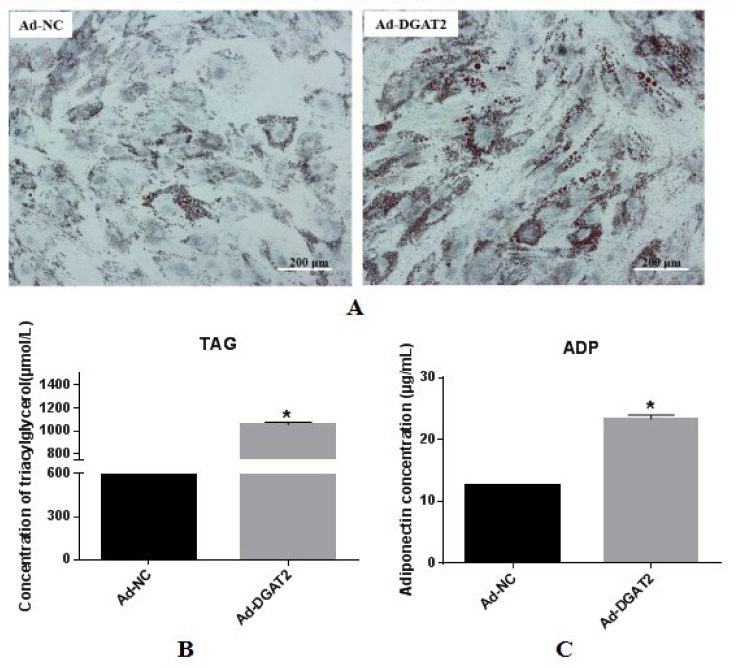
Overexpression of DGAT2 promotes BSCs lipid droplet accumulation, cellular TAG concentration, and adiponectin content in the medium. (**A**) Oil red O staining (scale bar: 200 μm). (**B**) Cellular TAG content after overexpression of DGAT2. (**C**) ADP content in the medium after overexpression of DGAT2. Values are presented as means ± SEM from three individual cultures. * *p* < 0.05.

**Figure 4 animals-12-01847-f004:**
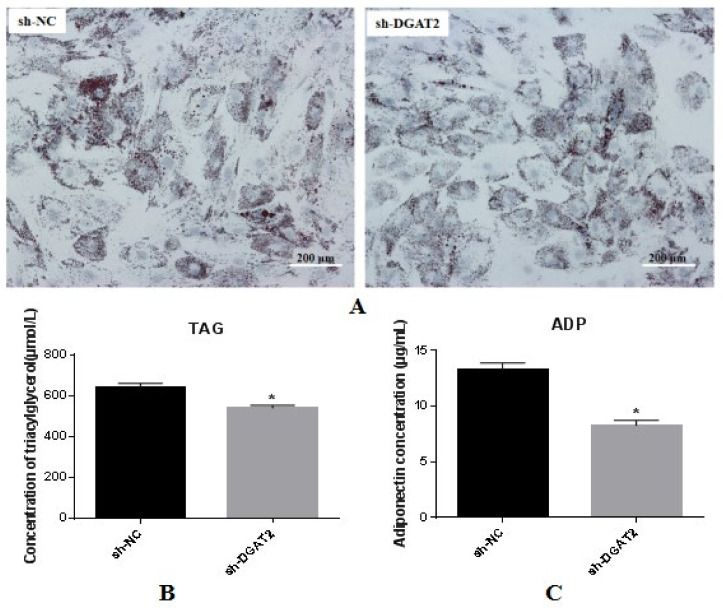
Interference of DGAT2 inhibits BSCs lipid droplet accumulation and reduces cellular TAG concentration and adiponectin content in the medium. (**A**) Oil red O staining (scale bar: 200 μm). (**B**) The effect of interfering DGAT2 on TAG content in BSCs. (**C**) The effect of interfering DGAT2 on ADP content in the medium. Values are presented as means ± SEM from three individual cultures. * *p* < 0.05.

**Figure 5 animals-12-01847-f005:**
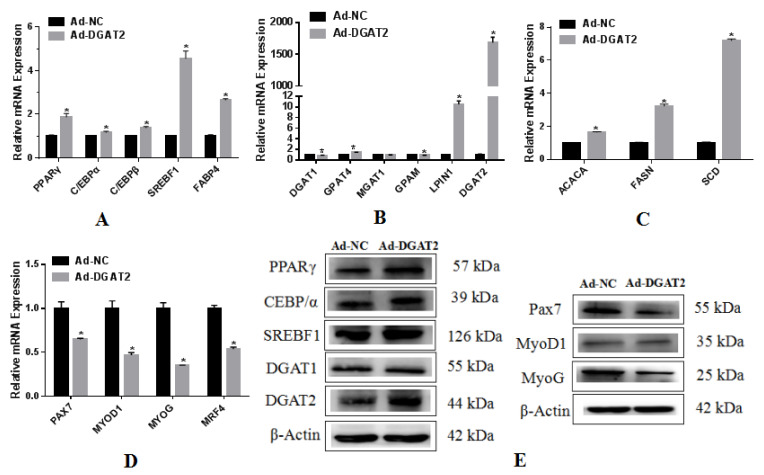
Overexpression of DGAT2 upregulates the expression of genes related to adipogenesis and lipid metabolism in BSCs. The mRNA expression levels of genes associated with (**A**) adipogenesis, (**B**) triacylglycerol synthesis, and (**C**) fatty acid synthesis in BSCs. Values are presented as means ± SEM from three individual cultures. * *p* < 0.05. (**D**) Overexpression of DGAT2 downregulates myogenesis-related gene expression in BSC. (**E**) Protein levels of adipogenesis markers PPARγ, SREBF1, C/EBPα, DGAT2, and DGAT1 and myogenesis markers Pax7, MyoD1, and MyoG.

**Figure 6 animals-12-01847-f006:**
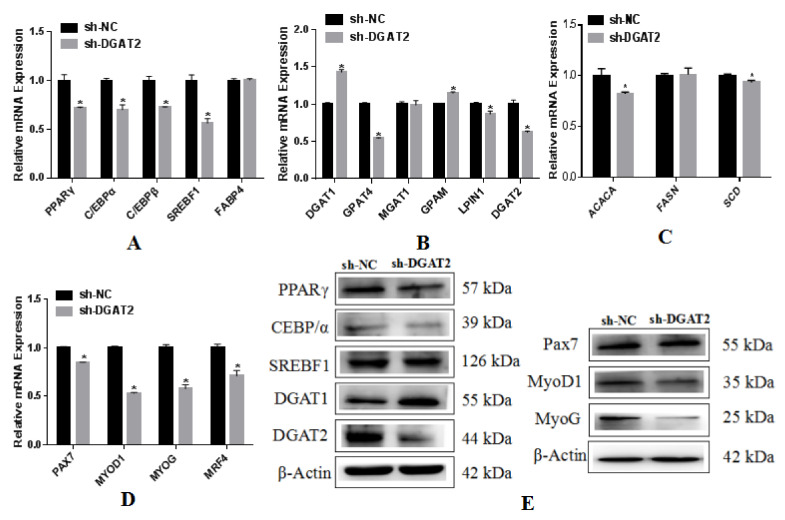
Interference with the expression of DGAT2 inhibits the expression of genes related to adipogenesis and lipid metabolism in BSCs. mRNA expression levels of genes associated with (**A**) lipogenesis, (**B**) triglyceride synthesis, and (**C**) fatty acid synthesis in BSCs. (**D**) Interference with DGAT2 downregulates myogenesis-related gene expression in BSC. Values are presented as means ± SEM from three individual cultures. * *p* < 0.05. (**E**) Protein levels of adipogenesis markers PPARγ, SREBF1, C/EBPα, DGAT2, and DGAT1, and myogenesis markers Pax7, MyoD1, and MyoG.

**Figure 7 animals-12-01847-f007:**
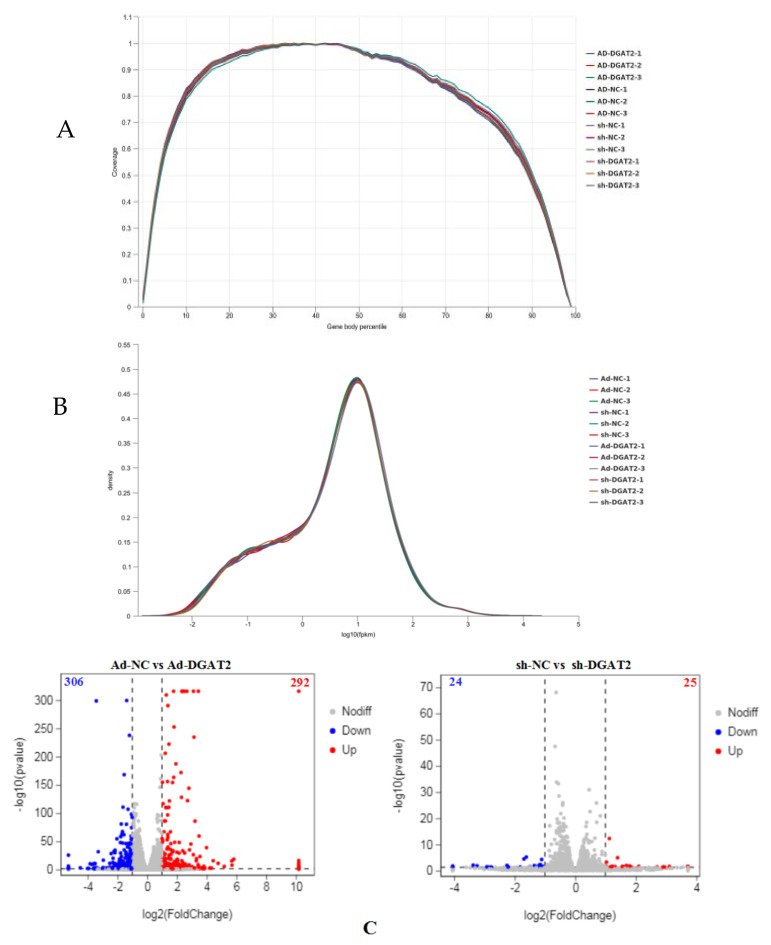
Screening of differentially expressed genes (DEGs) in BSCs infected with Ad-DGAT2/sh-DGAT2. BSCs were infected with Ad-NC, Ad-DGAT2, sh-NC, or sh-DGAT2 for 24 h; differentiation was induced with OA treatment for 96 h; and total RNA was extracted for transcription and sequencing analysis. (**A**) Gene coverage uniformity. (**B**) FPKM density distribution. (**C**) Volcano plot of DEGs.

**Figure 8 animals-12-01847-f008:**
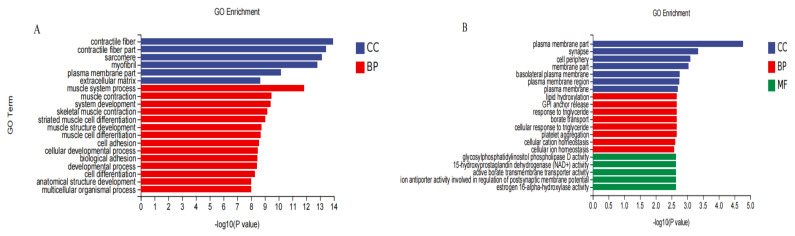
Gene ontology (GO) functional annotation and classification of differentially expressed genes (DEGs). (**A**) GO functional classification (Ad-DGAT2). (**B**) GO functional classification (sh-DGAT2). CC (cellular compound), BP (biological process), MF (molecular function).

**Figure 9 animals-12-01847-f009:**
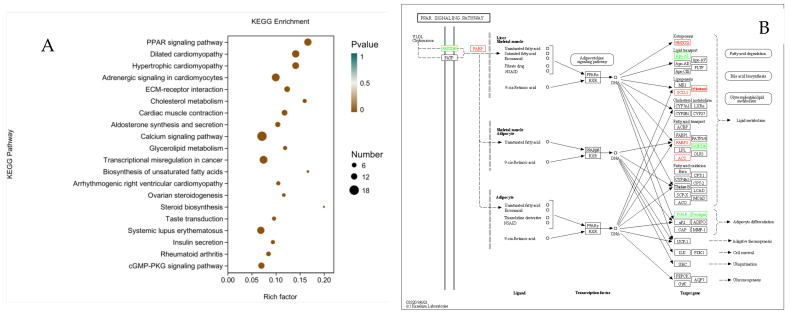
KEGG enrichment analysis of differentially expressed genes (DEGs) after overexpression of DGAT2. (**A**) KEGG enrichment factor map. (**B**) KEGG functional annotation of DEGs involved in the PPAR signaling pathway. (**C**) KEGG functional annotation of DEGs involved in the glycerophospholipid metabolic pathway. (**D**) KEGG functional annotation of DEGs involved in the biosynthetic pathway of unsaturated fatty acids. Red rectangles indicate DEGs significantly upregulated (log fold change > 1, false discovery rate (FDR) < 0.001); green rectangles indicate DEGs significantly downregulated (log fold change > 1, FDR < 0.001) in BSCs infected with Ad-DGAT2 compared with control Ad-NC.

**Figure 10 animals-12-01847-f010:**
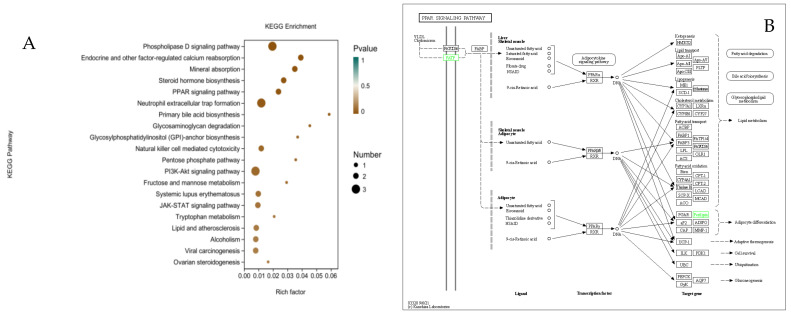
KEGG enrichment analysis of differentially expressed genes (DEGs) after interference of DGAT2. (**A**) KEGG enrichment factor map. (**B**) KEGG functional annotation of DEGs involved in the PPAR signaling pathway. Green rectangles indicate that DEGs were significantly downregulated (log fold change > 1, FDR < 0.001) in BSCs infected with sh-DGAT2 compared to control sh-NC.

**Table 1 animals-12-01847-t001:** Oligonucleotide sequences of sh-RNAs for the DGAT2 gene.

shRNA	Sense Strand	Anti-Sense Strand	Target Sequence
shRNA-108	AATTCGGTAGAGAAGCAGCTCCAAGTTTCAAGAGAACTTGGAGCTGCTTCTCTACCTTTTTTG	GATCCAAAAAAGGTAGAGAAGCAGCTCCAAGTTCTCTTGAAACTTGGAGCTGCTTCTCTACCG	GGTAGAGAAGCAGCTCCAAGT
shRNA-320	AATTCGCTACTTTCGAGACTACTTTCTTCAAGAGAGAAAGTAGTCTCGAAAGTAGCTTTTTTG	GATCCAAAAAAGCTACTTTCGAGACTACTTTCTCTCTTGAAGAAAGTAGTCTCGAAAGTAGCG	GCTACTTTCGAGACTACTTTC
shRNA-687	AATTCGCGCAATCGCAAGGGCTTTGTTTCAAGAGAACAAAGCCCTTGCGATTGCGCTTTTTTG	GATCCAAAAAAGCGCAATCGCAAGGGCTTTGTTCTCTTGAAACAAAGCCCTTGCGATTGCGCG	GCGCAATCGCAAGGGCTTTGT
shRNA-NC	AATTCGTTCTCCGAACGTGTCACGTTTCAAGAGAACGTGACACGTTCGGAGAACTTTTTTG	GATCCAAAAAAGTTCTCCGAACGTGTCACGTTCTCTTGAAACGTGACACGTTCGGAGAACG	TTCTCCGAACGTGTCACGT

**Table 2 animals-12-01847-t002:** Primers used for quantitative real-rime PCR.

Gene Symbol	Gene ID	Primer Sequence (5′–3′)	Length (bp)
*PPARγ*	NM_181024	F:ATCTGCTGCAAGCCTTGGA R:TGGAGCAGCTTGGCAAAGA	138
*SREBF1*	NM_001113302	F:CTGCTGACCGACATAGAAGACAT R:GTAGGGCGGGTCAAACAGG	81
*C/EBPα*	NM_176784	F:CCAGAAGAAGGTGGAGCAACTG R:TCGGGCAGCGTCTTGAAC	69
*C/EBPβ*	NM_176788	F:CAACCTGGAGACGCAGCACAAG R:CGGAGGAGGCGAGCAGAGG	143
*FABP4*	NM_174314	F:GTGGGCTTTGCTACCAGGAA R:GTGACCACACCCCCATTCAA	77
*DGAT1*	NM_174693	F:CTACACCATCCTCTTCCTCAAG R:AGTAGTAGAGATCGCGGTAGGTC	176
*DGAT2*	NM_205793	F:GACCCTCATAGCCTCCTACTCC R:GACCCATTGTAGCACCGAGATGAC	145
*GPAT4*	NM_001083669	F:AAGCAAGTTGCCCATCCTCA R:AAACTGTGGCTCCAATTTCGA	101
*GPAM*	NM_001012282	F:GCGAACAACTGGGAAAACCC R:GGCAACAATGCTTGCTCCAA	193
*LPIN1*	NM_001206156	F:AGTCCTCGCCACACAAGATG R:AGATGCCCTGACCAGTGTTG	137
*MGAT1*	NM_001015653	F:AGCCGTGGTGGTAGAGGATGATC R:TGCTCCTTGCCATTGTCGTTCC	132
*SCD*	NM_173959	F:TGCCCACCACAAGTTTTCAG R:GCCAACCCACGTGAGAGAAG	80
*ACACA*	NM_174224	F: GCCAAACCTCTGGAGCTGAA R: CGAGCTTCACCAGGTTGCTA	97
*FASN*	NM_001012669	F:CGCTTGCTGCTGGAGGTCAC R:GGTCTCAGGGTCTCGGCTCAG	141
*MYOD1*	NM_001040478	F:CCGACGGCATGATGGACTA R:CTCGCTGTAGTAAGTGCGGT	80
*MYOG*	NM_001111325	F:CAGTGAATGCAGCTCCCATAG R:GCAGATGATCCCCTGGGTTG	87
*MRF4*	NM_181811	F:TGGACCCCTTCAGCTACAGA R:ATGCTTGTCCCTCCTTCCTTG	139
*PAX7*	XM_015460690	F:TGCCCTCAGTGAGTTCGATT R:CGGGTTCTGACTCCACATCT	180

## Data Availability

The datasets generated and/or analyzed during the conduct of the study are included in this published article. Upon reasonable request, the datasets of this study are available from the corresponding author.

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
