# Peer review of "Overexpression of DGAT2 Stimulates Lipid Droplet Formation and Triacylglycerol Accumulation in Bovine Satellite Cells"

_animals, 2022, doi:10.3390/ani12141847_

Round 1
Reviewer 1 Report
line 36: Define BSC. Go through manuscript making sure that all abbreviations are adequately defined.
line 36: Insert and before SREBF1.
lines 85-91: Please consider stating a hypothesis for your research.
Table 1: Sequence data for shRNA-NC seems necessary.
line 127: What is source of polybrene?
line 189: What is composition of PBS? Source of RIPA lysis buffer?
Figure 4: What is the significance of TAG in the medium. I was not aware that BSCs secreted TAG.
Figure 5: Seems like beta actin expression are different in 5E.
line 364: Figureure?
Figures 7 and 8: Printing is too small for good readability. Enlarge?
Figure 8: Add A to first Figure 8 notation.
lines 466-468: Significance of SCFA reference is confusing in that beef TAGs do not contain SCFA.
line 506: This what?
References: Make sure capitalizations of titles are consistent.
Reviewer 2 Report
In this manuscript, Zhang and colleagues look at the effect of DGAT2 over expression on the adipogenic potential of bovine satellite cells. The study was well-conducted, for the most part, and interesting. I did notice some issues that the authors need to address before this manuscript is acceptable for publication:
The simple summary contains far too many acronyms (IMF, DEG, BSC, etc). Please summarize the findings more generically and in a way that makes this paragraph intelligible without having to read the manuscript.
A more thorough discussion of the adipogenic capacity of satellite cells is warranted. The authors describe this process as both "transdifferentiation" and "differentiation". While I appreciate that the plasticity/potency of satellite cells is an unresolved issue in the field, it would help readers of this article to explain this in a more clear way. The authors already have excellent references to this effect (2, 3, 22).
The header for section 2.3 reads like a result, not a method. This should be revised.
Statistical analysis throughout the paper should be revised. First off, the authors state in the Methods section that significance is set at P < .05 (line 263) and this should be followed throughout the manuscript. While I can appreciate a low P-value, the results can't be "more significant" so including ** for P < .01 is unnecessary. The authors include both * and ** in all figure legends, even in cases where letters are used to denote statistical differences (Fig 2B), or where only e.g., ** is shown (Fig 3). Please revise and consider using a single * for significantly different values.
Furthermore, the statistical analysis of the RNA-seq seems suspect as there is no mention of correcting for multiple comparisons (line 234). This is essential as transcript-wide testing will produce many false positives. The volcano plots seem to suggest low P values, as does the L2FC cutoff of 1.
Figure 7, panels C and D should be combined. Simply put the number from the histogram (C) in the appropriate quadrant of the volcano plot (D).
The figure quality is very low. Please provide high quality figures for review.
Figure 8 can not be split in this way. The authors need to combine panels (preferably) or make 8B and 8C figures 9 and 10.
Line 502 in the discussion mentions details the loss of myogenic capacity with all experimental manipulations. I feel like this should be more fully addressed as it is not clear that this is truly an adipogenic transdifferentiation or merely lipid accumulation. Related to the first comment, above.
